

# A coupled modelling system to assess the effect of Mediterranean storms under climate change

Riccardo A. Mel[1,2], Teresa Lo Feudo[3], Massimo Miceli[1], Salvatore Sinopoli[1,2], Mario Maiolo[1,2]

[1] Department of Environmental Engineering, University of Calabria, Arcavacata di Rende (CS), 87036, Italy
[2] Capo Tirone Experimental Marine Station, University of Calabria, Arcavacata di Rende (CS), 87036, Italy
[3] Institute of Atmospheric Sciences and Climate (CNR-ISAC), Lamezia Terme (CZ), 88046, Italy

*Correspondence to*: Riccardo Alvise Mel (riccardo_alvise.mel@unipd.it)

**Abstract.** Climate change will have an undeniable influence on coastal areas. In the last decades, the impact of storm surges has promoted multiple mitigation and adaptation strategies worldwide, including more robust sea defenses, development of 10 integrated modelling chains and warning systems, and improved storm impact management. However, as climate change seems likely to result in increased rates of both sea level rise and storm-related impacts, it is crucial to estimate the local probable extreme sea wave conditions, to properly reproduce the wave and hydrodynamic inshore field, and to investigate the effectiveness of sea defenses under different sea level rise scenarios. This work describes the first steps towards an innovative fully coupled modelling system composed of a hydrodynamic (2DEF) and wind wave model (SWAN). The 15 models are two-way coupled at half-hourly intervals exchanging the following fields: 2D sea level, surface currents and bottom elevation are transferred from 2DEF to SWAN; wave climate computed by SWAN is then passed to 2DEF by modifying the radiation stress. Numerical simulations have been performed to identify the impact of extreme storms at Calabaia beach, Italy, by combining sea level rise and extreme wave projections with the most recent georeferenced territorial data.

## 1 Introduction

Coastal areas contain a wide amount of life, supplying an estimated 43% of the world's ecosystem services (Jorve et al. 2014), and providing social, economic, and environmental benefits to the growing world population (Costanza et al. 1997; OECD, 2017). The increasing frequency of extreme flood events driven by climate change is crucial in actual urban planning research, as they represent one of the major challenges that the global risk society should face (Beck, 2013). Climate change 25 drives potential future sea hazards, as the greenhouse effect is expected to lead to global warming (IPCC, 2013). The resulting variation in the dynamic of atmospheric processes may cause further modifications in near-surface wind and pressure patterns, affecting storm surges, extreme wave conditions, and flooding, leading to significant alterations in coastal hydrodynamics (Casas-Prat and Sierra, 2010; Labuz et al., 2015). Weisse and von Storch (2010) provided an overview of the state of the art concerning the relationship between anthropogenic climate change and wave climate. The relevance of the 30 topic and the consequent claim of new paradigms for the sustainable transformation of the threatened coastal regions is also





recognized in the Global Agenda for Sustainable Development 2030, which recommends making cities and human settlements inclusive, safe, long-lasting, and sustainable (Obj. 11), and urges that policy makers, stakeholders, and authorities face the consequences of climate change (Obj. 13) (Mariano et al., 2021).

Coastal dynamics are driven by the interaction of different geospheres over a wide range of timescales. The main factors triggering changes in coastal areas are geological, geomorphological, hydrodynamic, biological, climatic, and anthropogenic (Labuz et al., 2015). Sea storms are one of the most common extreme events that affect coastal areas, characterized by strong winds, waves, and increase in sea level (Morton et al., 2011). Sea waves, caused by the effect of local wind climate, are often superimposed on swell waves moving inshore from a distance. Interaction between sea and swell waves can cause unpredictably high waves. Extreme sea stormy conditions are especially important, as they impinge violently on the

shoreline and may result in extensive damage to the local population and assets as a consequence of extremely high-water level, flooding and erosion processes (Katoh and Yanagishima,1988; De Zolt et al., 2006; Kortekaas and Dawson, 2007; Soomere et al., 2008; Switzer, 2008; Switzer and Jones, 2008; Lario et al., 2010; Rodríguez-Ramírez et al., 2015). These events affect the morphologic evolution of the shoreline by driving a series of morphodynamic responses proportional to the energy of the storm, with significant consequences on coastal geomorphology and a general enhance of the existing retreat

rates (Morton and Sallenger, 2003). Furthermore, empirical evidence shows a significant impact of extreme events on coastal settlements and territories, and for the consequent economic, social, and environmental damages (Mariano et al., 2021). Their effects may vary from minor erosion and over-wash of the shoreline to the complete devastation of coastal settlements, threatening the coastal communities. In the short term, sea storms can devastate crops, infrastructures, and take the lives of humans and livestock. In the long term, they may trigger significant erosion processes and losses of wide areas of land (Van

Gelder et al., 2000). The seasonal fluctuation of these events depends on the number and intensity of the storms during a particular year (Dewall, 1979) and the energy content related to the morphodynamic responses (Edelman, 1972; Paladini de Mendoza et al., 2014). Among the multiple threat caused by climate change, coastal flooding in urban areas is one of the most critical due to the projected global sea level rise (SLR). Coastal flooding originates from multiple non-linear factors (e.g., waves, storm surges, rain, hydrological runoff), combined with SLR and interacting across multiple space and time

scales (Mariano et al., 2021). Wave storms are the principal driver of short-term coastal erosion and flooding; hence it is important to understand their occurrence and to properly model the nearshore hydrodynamic in view of climate change (Besio et al., 2017). This is particularly significant in case of micro-tidal environments, such as the Mediterranean Sea, where extreme events are expected to be superimposed to SLR scenarios, exacerbating the flooding hazard even in the case of a possible storminess reduction (Androulidakis et al., 2015; Gaeta et al., 2018).

Although the wave climate in the Mediterranean Sea has been extensively addressed, regional differences are significant. Sartini et al. (2015) linked the seasonal wave climate to the local ground pressure field, strongly suggesting that the local typical meteorological conditions produce different temporal regimes of storm waves (see also Besio et al., 2017). Mendoza et al. (2011) identified three main synoptic situations over the Mediterranean Sea: (i) the Mediterranean Cyclone, characterized by a pressure trough over the Mediterranean Sea, which may cover a wide area with a strong intensity


(Radinovic, 1987; Campins et al., 2000; Trigo et al., 2002); (ii) the south advection, characterized by a pressure trough over the Atlantic Ocean which produces large ground pressure gradients and subject the Mediterranean Sea to the action of southern winds; (iii) the east advection, characterized by the a high pressure center over North-Central Europe and a low pressure trough over North Africa, resulting in eastern winds. However, Mendoza et al. (2011) did not find a clear relationship between storm power and wave energy content, suggesting that other storm parameters must be taken in

account, such as the position of the pressure trough and the track of the storm. Wave climate trend in the Mediterranean Sea has been extensively analyzed in literature. Lionello and Sanna (2005) performed a statistical analysis of the wave climate for the last decades of the 20th century, showing no significant long-term trends. Pomaro et al. (2017) analyzed the trends in significant wave height in the Northern Adriatic Sea in the period 1979–2015, showing a clear reduction of the highest percentile (99th) of significant wave height and a smaller growth of the 50th and 75th ones. Caloiero et al., (2019) and Lo

Feudo et al. (2022) found no significant long - term trends for the Tyrrhenian Sea.

Long-term wave statistic usually consists in field data achieved within a long period of time or dataset computed by means of mathematical models. Although the estimation of extreme values is crucial in risk-based design and operation of sea structures (Bitner-Gregersen, 2015; Bitner-Gregersen et al., 2016), there are different techniques to employ the data and to fit and optimize the theoretical distribution (Van Vledder et al., 1993; Bouws et al., 1998; Ferreira and Guedes Soares, 1999;

Guades Soares and Scotto, 2001; Prpic-Orsic et al., 2007; Vanem, 2015; Orimolade et al., 2016). Extreme values distributions include the Gumbel, Log-Pearson, log-normal and Weibull distributions. Jeong et al. (2008) suggested that Gumbel and Weibull distributions are the most suitable for extreme wave analysis (see also Park et al., 2020). Extreme sea storms belong to interdepended multidimensional variable (Liu et al., 2019). The estimation of probable extreme sea wave conditions is fundamental for the design of maritime works, improving the knowledge on the effects of ocean processes on

the seasonal variability of the sediment transport rate or the influence of wave exposure on deposition and erosion rates (Sartini et al., 2015). In this context, the development of efficient tools to reproduce the inshore wave and flow dynamics is fundamental for actual operational and forecasting applications in coastal areas (Bonaldo et al., 2018). In particular, the support of two-dimensional mathematical modelling is crucial, since extreme value distributions of single variables cannot effectively meet the need of researchers (Liu et al., 2015; Wang et. al., 2017; Chen et al., 2018). High-resolution models

support planning and decision processes in coastal areas, reproducing the typical local features at a coastal engineering scale (e.g., nonlinear processes of wave propagation and their interactions with coastal structures). Thus, the development of multipurpose mitigation measures, together with the increasing of coastal defense efficiency, requires a challenging prediction of the inshore wave climate and hydrodynamic (Gaeta et al., 2018). The interaction between wind, wave, and hydrodynamic fields, which control the momentum and energy exchange between the atmosphere and ocean, need to be

properly investigated. Sea level and surface currents are driven by sea state, which depends in turn on the local hydrodynamic. These complex feedback mechanisms can be modelled by coupling wind-wave and hydrodynamic modules which to date have been developed separately (Clementi et al., 2017). Jonsson (1990) and Cavaleri et al. (2012) provided a coupling overview achieved at various levels of complexity.





The aim of this study is to identify the extreme sea wave conditions affecting in the southern Tyrrhenian Sea (Italy) using the

Gumbel distribution, and to simulate their nearshore effects at Calabaia beach, with and without the perched nourishment carried out in 2006 (Maiolo et al., 2020a), and adding the possible effect of SLR. Wave data for the period 1950–2019 were obtained from the global atmospheric reanalysis ERA-Interim by the European Centre for Medium-Range Weather Forecasts (ECMWF). The most important recent storm which affected the Mediterranean Sea (25 – 29 December 1999) has been reproduced by coupling, for the first time, the wind-wave model SWAN to the hydrodynamical model 2DEF. Further

simulations investigated the coastal hydrodynamic under steady conditions for different offshore wave characteristics. The coupled modelling system is composed of the hydrodynamic model 2DEF (Defina et al.,1994; Defina, 2000) and the third-generation wave model Simulating Waves Nearshore (SWAN, see Stopa et al, 2011; Garcia-Medina et al., 2021). The performance of the coupled wind-wave and hydrodynamic models used in the present work is assessed by comparing numerical results to the same outcomes computed by means of the MIKE 21-3 Coupled Model FM (MIKE, see Danish

Hydraulics Institute, 2007), for inshore significant wave height, wind-wave setup, sea level and surface currents. The bottom elevation of the area has been obtained through a specific high-resolution topographic survey performed by means of an Unmanned Aerial Vehicle (UAV) system, which adopts a photogrammetric technique based on the innovative Structure from Motion (SfM) algorithm. We extrapolated accurate 3D and 2.5D data, with a significant improvement in terms of resolution with respect to traditional satellite images. Furthermore, digital models achieved from proximity photogrammetry

have no limitations in extrapolating local vectors (e.g., profiles, sections, contours, point grids, etc.).

The paper is organized as follows. In Section 2, we describe the characteristics of the Calabaia beach, briefly presenting the methodology, the dataset, and the implementation of the modelling system. The estimation of probable extreme sea wave conditions is presented in Section 3, together with the reproduction of the storm effects under different SLR and morphology scenarios. A set of conclusions closes the paper.

## 2 Material and Methods

### 2.1 The study area

Calabria region is located in south of Italy, ranging between 37°55' and 40° latitude North and between 15° 30' and 17° 15' longitude East. The western part of the region is bounded by the southern Tyrrhenian Sea, while the eastern, southern and eastern sides are bounded by the Ionian Sea (Fig. 1). Apennines run along the whole region from north to south, consisting of

five main ranges, namely, Pollino, Catena Costiera, Sila, Serre, and Aspromonte, characterized by peaks heights between 1,500 m and 2,000 m (Federico and Bellecci, 2004).

The Calabaia beach is located into the municipality of Belvedere Marittimo (Province of Cosenza, see Fig. 1). The morphology of the shoreline is mostly characterized by metamorphic and sedimentary rocks directly extending to the sea and narrow coastal plains (Maiolo et al., 2020a). The Tyrrhenian coastal areas of Calabria region are densely populated,

providing multiple services to the local population and economy, which is mostly based on fishing, tourism, and leisure

activities (Maiolo et al. 2020a). However, large part of the shoreline is threatened by flooding and erosion processes driven by the major sea storms (Foti et al. 2019). In the last decades, the Southern Tyrrhenian shoreline has been intensively urbanized, enhancing erosion processes and requiring the building of multiple flood defences (Greco, 1994; Warnken et al., 2018). At the beginning of the 2000s, the large loss of sediment in the municipality of Belvedere Marittimo led to the

restoration of the Calabaia beach by means of a complex intervention made of (i) a submerged breakwater located 250 m seaward, 2.5 m below the mean sea level and 700 m long; (ii) multiple semi-submerged groynes connecting the heads of the submerged breakwater to the shoreline; (iii) a beach nourishment (see Maiolo et al., 2020a and 2020b for more technical details about). A further monitoring campaign proved the effectiveness of the intervention, as the shoreline has reached the equilibrium profile (Regione Calabria, 2010). Prevailing wind and wave directions are from the third and fourth quadrant

(Lo Feudo et al., 2022). The mean spring tidal range is about 0.5 m, with a negligible interaction between tidal currents and wave currents.

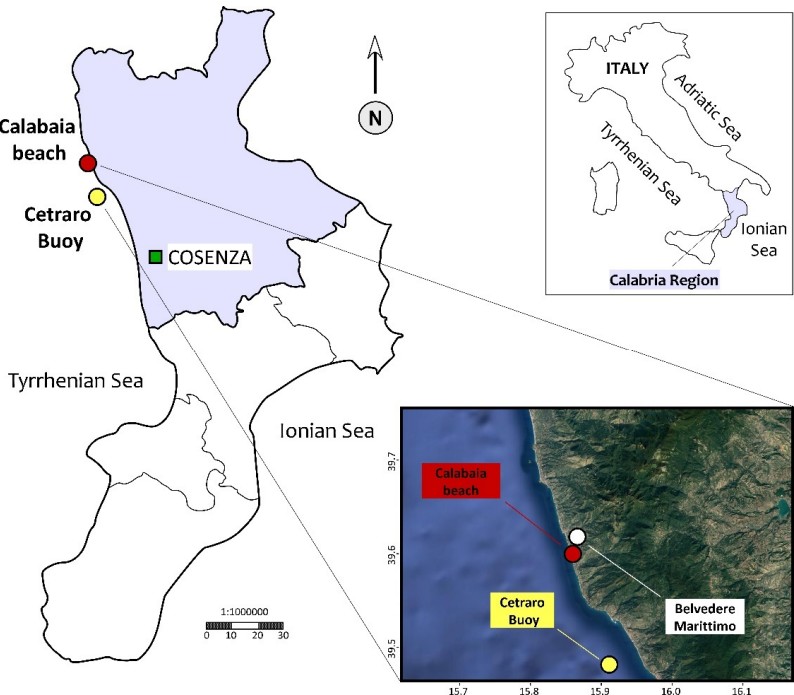

**Figure 1.** Calabria region and location of the Calabaia beach and the Cetraro buoy (© Google Maps 2020).

## 2.2 ERA 5 wave dataset

Wave climate statistic has been computed through ERA 5 hindcast reanalysis produced by ECMWF and extended from January 1950 to December 2019, with a temporal resolution of six hours (see Hersbach and Dee 2016; Hersbach et al. 2018; Lima et al. 2018; Ramon et al. 2019 for more technical details). The exceptional storm of 25 – 29 December 1999 has been reproduced forcing the coupled SWAN+2DEF and MIKE models with the data collected by a Datawell Waverider buoy located at Cetraro (39°27′12″ °N 15°55′06″ °E, 100m depth, see Fig. 1), belonging to the Italian National Sea Wave Measurement Network (RON, see Bencivenga et al. 2012). Although the actual performance of wind-wave models is generally good, for closed basins (i.e., the Mediterranean Sea) winds forcing is generally underestimated, with a significant impact on wave modelling due to the lack of knowledge of detailed physiographic features. An extensive comparison between modelled wave height using ECMWF wind fields and buoy data shows an underestimation of wave heights of almost 25% and of wave periods of 5 – 10 % (Fig. 2, see Lo Feudo et al., 2022 for a thorough analysis). Thus, the extreme wave climate analysis (Section 3.1) has been performed by correcting the ERA 5 dataset accordingly (i.e., by dividing the wave height and period by factors equal to 0.74 and 0.93 respectively, hereinafter named as ERA 5 normalized dataset).

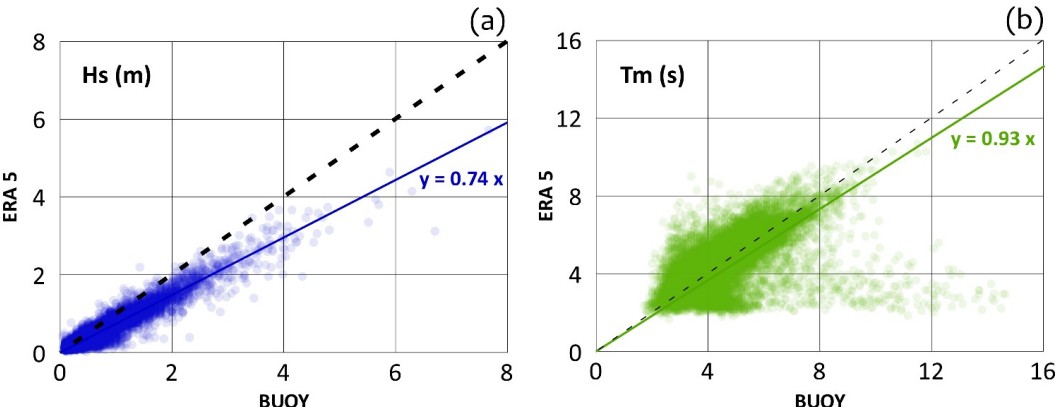

**Figure 2.** Dataset 1950 - 2019, time step of 6 hours. Scatter plot ERA 5 versus buoy dataset at Cetraro. Panel (a) shows the significant wave height (Hs), panel (b) the mean period (Tm).

## 2.3 The model system

The third generation of the SWAN wave model has been developed by Delft University of Technology and it is widely used for high resolution offshore and near shore wave predictions, including the surf zone (Holthuijsen et al., 1993; Ris et al., 1994). SWAN is a fully discrete spectral model based on the wave action balance equation, driven by boundary conditions and local winds, and implicitly considering the interaction between waves and currents through radiation stresses (Phillips,

1977; Booij et al., 1996). The implicit numerical propagation scheme significantly reduces the computational effort in shallow waters. It adopts a Eulerian formulation of the discrete spectral balance of action density that accounts for refractive propagation over arbitrary bathymetry and current fields (Booij et al., 1999). Wind generation, depth-induced wave breaking,

white-capping, bottom dissipation, triad, and quadruplet wave-wave interactions are represented explicitly. SWAN outcomes agree well with analytical solutions, laboratory, and observations (Booij et al., 1999).

The 2DEF hydrodynamical model solves the full 2D shallow water equations on unstructured triangular grids through a semi-implicit staggered finite-element method, based on mixed Eulerian-Lagrangian approach (Defina, 2003). The 2DEF model adopts a statistical sub-grid approach for bottom elevations (Defina et al.,1994; Defina, 2000), achieving a physically

based, accurate and stable treatment of wetting and drying processes (D'Alpaos and Defina, 2007). The Boussinesq approximation (Stansby, 2003) has been adopted to determine the depth-integrated horizontal dispersion stresses, whereas the eddy viscosity is solved according to Uittenbogaard and van Vossen (2004). The 2DEF model has been extensively used in river engineering (e.g., Martini et al., 2004; Viero et al., 2013; Mel et al., 2020). In recent years, 2DEF has been coupled with a wind-wave module solving the wave action conservation equation parameterized using the zero-order moment of the

wave action spectrum in the frequency domain (named as WWTM, see Carniello et al., 2005). WWTM has been extensively used in Venice Lagoon and in other shallow coastal and transitional semi-closed water bodies (e.g., Mariotti et al., 2010; Zarzuelo et al., 2018; Mel 2021; Mel et al., 2021). Notably, the simplifications adopted in the WWTM wind-wave module are not suitable to produce reliable open-sea analysis.

In this study, the 2DEF has been coupled with SWAN to investigate the nearshore hydrodynamic of the southern Tyrrhenian

Sea. The coupling is achieved through a half-hourly exchange of instantaneous fields of sea level, surface currents and bottom elevations from 2DEF to SWAN, whereas the radiation stress and the orbital wave velocity evaluated from SWAN is passed to 2DEF. SWAN uses a structured grid with rectangular elements, whereas 2DEF adopts an unstructured grid with triangular elements. The SWAN grid is included into the boundaries of the 2DEF domain. SWAN elements show similar size of the 2DEF grid, and their bottom elevation is based on an interpolation on the 2DEF grid.

Results achieved by the SWAN+2DEF coupled modelling system have been compared to those obtained by MIKE, an integrated modelling system based on an unstructured grid solving the 2-3D incompressible Reynolds averaged Navier-Stokes equations, under the hypothesis of Boussinesq and of hydrostatic pressure. Shallow water equations are solved by means of the approximate Riemann solver (Roe 1981; Jawahar et al. 2000); wave fields are described by the wave action conservation equations (Anastasiau and Chan 1997). The model is thoroughly described in the tutorial compiled by the

Danish Hydraulics Institute, 2007.

### 2.4 Model setup

The model grid is closed 4 km north of Diamante, 3 km south of Cape Bonifati and 7 km seaward (Fig. 3a). The 2DEF computational mesh consists of about 25,000 nodes and 50,000 triangular elements of characteristic size (side-length) of almost 500 m; smaller elements (up to 5 m) describe the inshore area and the urban settlement of Calabaia. The SWAN grid
density is cross-shaped. The shortest side of the rectangular elements faces the study area, where the elements are squared
with characteristic size of 6 m (Fig. 3a). The model is forced by imposing (refer to Fig. 3a): (i) wave climate at the seaward
boundary section (yellow dashed line) every 30 minutes; (ii) lateral conditions at the two lateral boundary sections (grey
lines); (iii) land condition at the coastline boundary section (black line), (iv) uniform wind field over the whole domain.

In this study we reproduced the storm occurred on 25 – 30 December 1999 together with some simulations under steady

conditions. Specifically, we reproduced seven wave directions (i.e., 165 °N; 195 °N; 225 °N; 255 °N; 285 °N; 315 °N; 345
°N) and, for each wave direction, three significant wave heights (i.e., 4 m; 6 m; 8 m). For each of the 22 simulations, we
reproduced five scenarios of sea level rise (i.e., +0.0 m; + 0.5 m; +1.0 m; +1.5 m; +2.0 m) and, in turn, three morphological
scenarios (i.e., present condition of the shoreline, after the intervention described in Section 2.1 and in Maiolo et al., 2020a
and 2020b; former condition, without such intervention; present condition without the submerged barrier and the semi-

submerged groynes), for a total of 330 simulations.

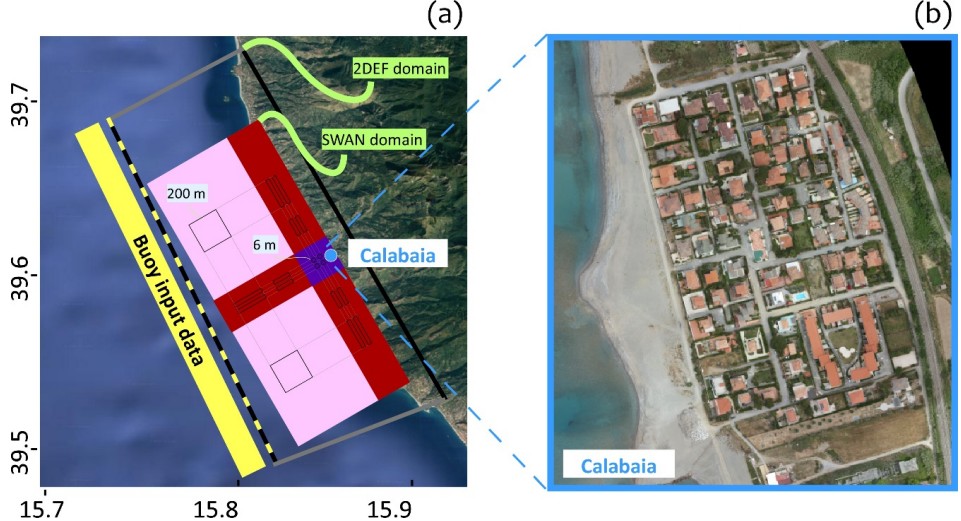

**Figure 3.** SWAN+2DEF domain: (a) yellow dashed line indicates the seaward boundary section where the model is forced by the wave
climate gauged at the Cetraro buoy; grey lines the two lateral boundary condition sections; black line the coastline boundary section. Blue
bullet refers to Calabaia beach (© Google Maps 2020). (b) high-resolution orthophoto of Calabaia beach, survey of 26[th] May 2021.

Topographic and bathymetric data have been collected during a specific survey (May 2021, see Fig. 3b and Fig. 4). For the
topographic survey, we used an UAV DJI Matrice 300 RTK (positioning accuracy 1.0 cm horizontal and 1.5 cm vertical),
equipped with Zenmuse P1 camera. The main characteristics of the camera are: sensor size 35.9 x 24 mm (full frame);
resolution 45 MP (pixel size: 4.4 µm); lens 35 mm F2.8; FOV 63.5°; aperture range f/2.8 – f/16. The total area covered by
the UAV was about 40 ha, flying at altitude of 120 m above the ground level. The 672 shots were post-processed with
photogrammetric software, which produced: dense point cloud (more than 150M points), orthophotos, digital surface model (DSM) and digital terrain model (DTM). The resolution of the orthophoto and DSM is 1.47 cm/pixel, the resolution of the DTM 7.35 cm/pixel. This dataset has been integrated by further bathymetric surveys performed by means of an echosounder (Lawrence Hook[2] v.9) mounted on a boat and pinging a beam of sound downward at the seafloor. These additional surveys have been performed between June and October 2021, for a total of about 1M points.

Results have been analyzed every one-hour (i.e., by using a moving average on the output printed every 15') in five sections perpendicular to the shoreline and equally spaced (100 m, see Fig. 4). In addition, the flow rate has been assessed between sections $Q_A$ and $Q_B$ (Fig. 4 black line). Bottom elevations ($z$) of the five sections are: - 2.0 m (section 1); - 3.4 m (section 2); - 5.9 m (section 3); - 9.3 (section 4); - 10.9 (section 5). Section $Q_A$ is located in correspondence to the submerged barrier ($z =$ - 2.5 m); section $Q_B$ in the ground above the mean sea level ($z = + 3.0$ m).

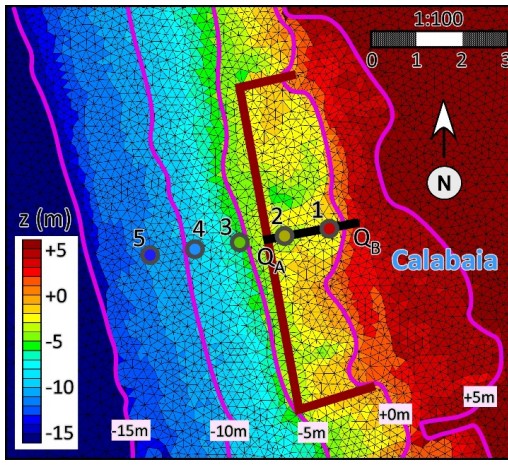


**Figure 4.** 2DEF grid and bathymetry at Calabaia. Brown line represents the submerged barrier. Red (section 1), yellow (section 2), green (section 3), light blue (section 4), blue (section 5) bullets represent the five output sections where wave characteristics and sea level have been compared. Black line $Q_A$ – $Q_B$ identifies the transect within the submerged barrier used to assess the longitudinal flow rate.

**2.5 The storm of 25 – 29 December 1999**

In December 1999, North Atlantic, Europe and Mediterranean Sea saw a series of heavy winter storms (namely Anatol, Lothar and Martin), which claimed more than 130 lives and caused about 13 billion Euros worth of total economic losses (Ulbrich et al., 2001). Their size and ability to travel over long distances and over land without weakening produced total losses comparable to hurricanes. On 26 December the storm Lothar hit the central Europe, followed by the storm Martin between 27 and 28 December which left great damage from north-western France to southern Germany, Switzerland, Spain,

and Italy. Extreme winds (wind gusts over 40 m/s, breaking several records) struck these countries, causing widespread


damage to the population and assets, multiple electricity breakdowns, and several indirect consequences, such as the disruption of computer networks, refrigeration plants, and lower earnings for many companies (Bründl and Rickli, 2002). The storms were characterized by extremely high baroclinicity near the cyclones track over the eastern North Atlantic, extending partly into Europe. Lothar originates from a strong divergence area developed between Brittany and Cornwall,

intensifying almost explosively into an extratropical cyclone of 300 km in diameter and with internal pressure gradients comparable to hurricanes of category 2. The pressure trough crossed the Normandy coast in the early hours of 26 December, raging later across northern France, Belgium, Germany, and only when nearly half-way across Poland, finally weakened. In the mid-afternoon of 27 December, a second cyclone, named as Martin, landed about 200 km south of Lothar's landfall, showing similar genesis and characteristics. Martin crossed rapidly central France, Switzerland, northern Italy, and then

weakened after approached the Balkans. Although these storms were characterized by extremely high-top wind speeds in the Mediterranean Sea and lowlands, the medium wind speeds were within the range expected for a strong storm. It should be noted that low-resolution climate models were unable to describe the trajectory and wind field of small-scale cyclones (see Ulbrich et al., 2001 for more detail of the characteristics of the events and their associated forecasts). As concerns the southern Tyrrhenian Sea, the windstorm produced by the cyclones Lothar and Martin is one of the most intense ever

recorded at Cetraro buoy. The first cyclone produced large winds over the central Mediterranean basin during 27 December, while the second storm was active on 28 and 29 December. Maximum wave height exceeded 10 m and large damages were reported along all the southern Tyrrhenian coast (see Federico and Bellecci, 2004 for more details about the events). Figure 5 shows surface sea level pressure field derived from CFS reanalysis on 26-29 December 1999, 12 UTC. Figure 6 illustrates wave height, direction, and period recorded at Cetraro buoy on 25-30 December 1999.


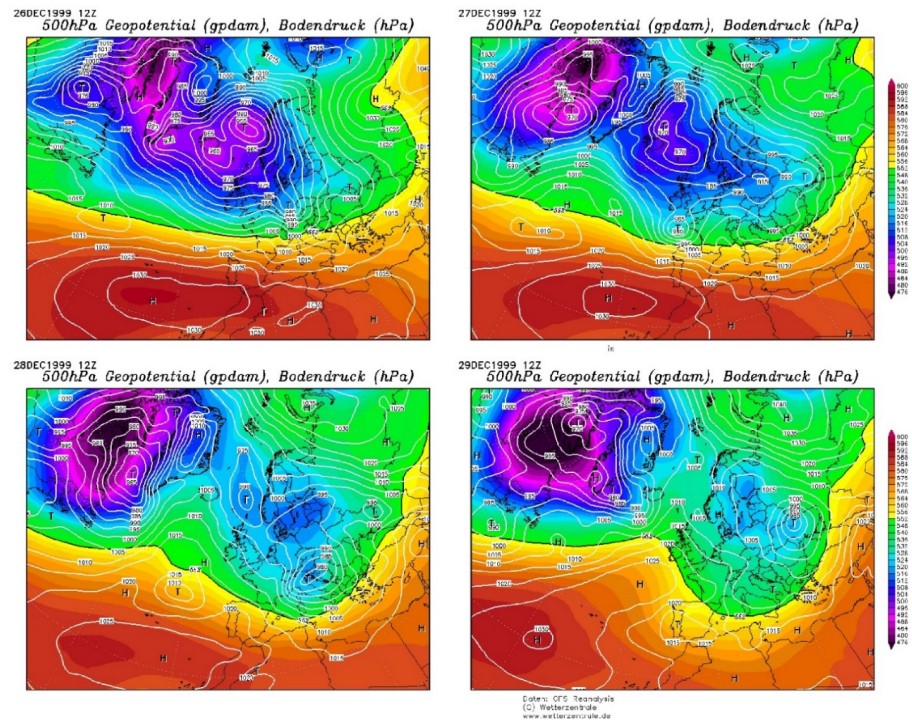


**Figure 5.** 26 – 29 December 1999, 12 UTC. CFS reanalysis of geopotential and ground pressure. Source www.wetterzentrale.de

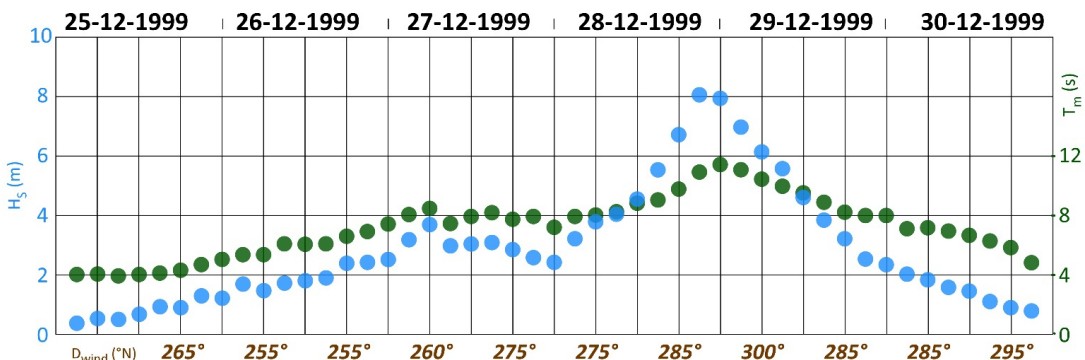

**Figure 6.** 25 – 30 December 1999. Significant wave height (blue bullets), mean wave period (green bullets), and wave direction (brown

labels) gauged at Cetraro Buoy.


### 2.6 Sea Level rise projections

The IPCC Sixth Assessment Report on climate change (AR6), published on 9 August 2021 assumes that SLR scenarios are likely to worsen the flood threat for worldwide coastal areas (Fig. 7). AR6 is based on the latest climate model data and analytical techniques to assess the impacts of climate change. Specifically, the most important difference between the previous IPCC report (AR5, see IPCC 2013) is the use of a new generation of climate models, implementing the latest science and technology to produce projections of future climate. More than 30 institutions contributed to over 40 models. AR6 is based on (i) five Shared Socioeconomic Pathways (SSPs), i.e., five scenarios of projected socioeconomic global changes up to 2100 related to greenhouse gas emissions under different climate policies (Armstrong et al., 2012; O'Neill et al., 2014; Dellink et al, 2015; Kc and Lutz, 2015; Riahi et al., 2017), and (ii) the Representative Concentration Pathway (RCP) scenarios, related to the greenhouse gas concentration trajectory, i.e., the possible radiative forcing in the year 2100, expressed in W/m$^2$. The SSP scenarios are named as *SSP1 Sustainability* (taking the green road), *SSP2 Reference scenario* (middle of the road), *SSP3 Regional rivalry* (a rocky road), *SSP4 inequality* (a road divided), *SSP5 fossil-fueled development* (taking the highway). Specifically, in the scenario *SSP1*, the world would gradually shift gradually toward a more sustainable path (i.e., sustainable development, reduction of inequalities, and lower use of the land resources). Scenario *SSP2* assumes steady social, economic, and technological trends, with respect to historical patterns. In the scenario *SSP3* a resurgent nationalism would shift the policies to become increasingly oriented toward national and regional security energetic and food issues, with a low interest for addressing environmental concerns. Scenario *SSP4* hypothesis are increasing inequalities and poorly educated societies, with a degradation of the social cohesion and environmental policies focusing on local issues only. In Scenario *SSP5*, the markets are assumed very competitive and integrated, with an exploitation of huge fossil fuel resources and the adoption of resource and energy intensive lifestyles. The RCP scenarios range from 1.9 W/m$^2$ (a pathway that achieves the aspirational goal of the Paris Agreement) and 8.5 W/m$^2$, assuming an increase of greenhouse emissions throughout the present century (IPCC, 2013).

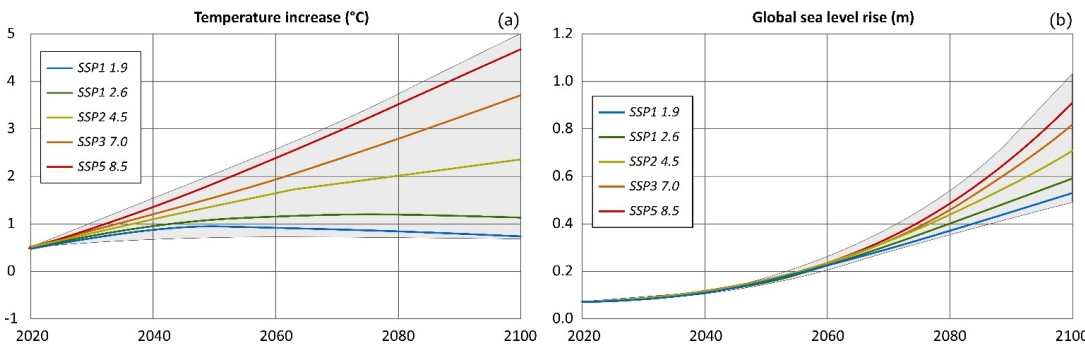



**Figure 7.** Temperature (a) and global sea level (b) projections according to the five likelihood scenarios provided by the IPCC 6th Report. The values refer to 1st January of each decade and are computed with respect to the period 1986 – 2005, shaded grey areas represent the total confidence interval.

Regional estimates of SLR are still rather uncertain (Rinaldo et al., 2008), as significant acceleration must occur in the present century to materialize the IPCC predictions (Tomasicchio et al., 2018). Frederikse et al., 2020 expect for the Mediterranean Sea possible discrepancies in relative SLR with respect to the three RCPs scenarios, with particular reference to the period 2050 – 2100, due to a possible reduction of the contribution of icesheet melting in the Subpolar North Atlantic basin and Greenland. This possible deviation has been widely addressed by means of mathematical models reproducing the circulation within the Mediterranean Sea. Adloff et al., 2018 and Slangen et al., 2017 estimate a (local) SLR in the Mediterranean Sea lower of 10% - 20% if compared to global SLR projections. In this study we assumed four different SLR projections for the year 2100: 0.5 m, corresponding to the most likely scenario; 1 m, corresponding to the upper range of the RCP 8.5 scenario; 1.5 m; 2 m, corresponding to the extreme scenario (not shown in Fig. 7b).

## 3 Results and discussion

### 3.1 Extreme wave analysis

Extreme wave climate study is based on data collection, selection, and analysis (Mathiesen et al., 1994). In this work, we processed the 1950 - 2019 ERA 5 normalized dataset (i.e., by applying the calibration factors computed through the comparison with the Cetraro buoy, see Section 2.2). We adopted the Gumbel distribution, analyzing the yearly maxima of significant/maximum wave height and mean/peak wave period, following the method described in Guedes Soares et al., 1996 (Fig. 8). For a return period (Tr) of one year, we computed a significant/maximum wave height of almost 4.5/9 m, and a mean/peak period of 10/11.5 s. For Tr = 10 years, we found a significant/maximum wave height of 6.5/13 m, and a mean/peak period of 11/13 s. For a 100-year return period, we computed a significant wave height of almost 8 m and a mean wave period of 12 s, similar to the values recorded on 28 December during the storm Martin, confirming its importance. For the same return period, we found a maximum wave height of 16 m and a peak period of 15 s (Fig. 8a,b). Notably, the maximum wave height is about double of the significant wave height and the peak period 1.15 times the mean period, independently of Tr.

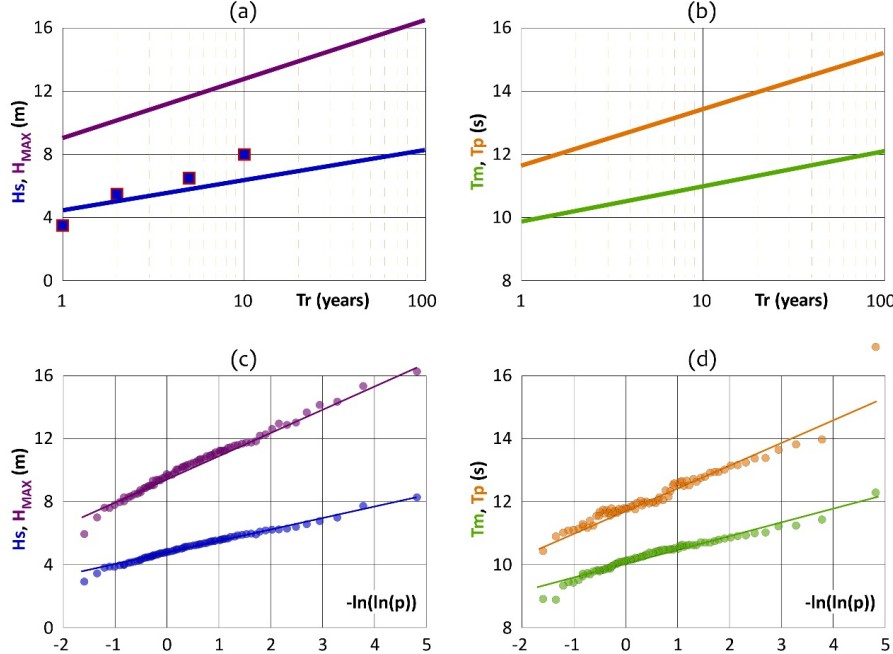


**Figure 8.** Significant (Hs) and maximum (H$_{MAX}$) wave height (a) and mean (Tm) and peak (Tp) wave period (b) estimated by means of the Gumbel distribution on the basis of the calibrated ERA 5 dataset (solid lines). Red-blue squares (a) represent the inverse of the Hs frequency gauged at the Cetraro buoy over the period 1999-2008. Panels (c) and (d) represent the same parameters of (a) and (b), but
showing the Gumbel best fit of the 70 values computed by using the least square method.

As the wave direction variability can affect the estimation of the extreme values, the assumption of the performed univariate analysis (Fig. 8) implies equal probability of extremes arriving from all directions during all seasons and wind patterns with such characteristics, leading to a significant simplification of the problem (Katalinic and Parunov, 2020). Specifically, the analysis could be refined by taking in account the dominant local wave direction patterns (see Lo Feudo et al., 2022) and
separating the calculation of the return time values accordingly. For the Tyrrhenian Sea, most extremes occur from north-western, west, and south-western winds that present different fetches, representing statistically separate datasets. Figure 9 shows the individual analysis of significant and maximum wave height extremes caused by distinguishable wind patterns of 30° range, to provide more accurate predictions.

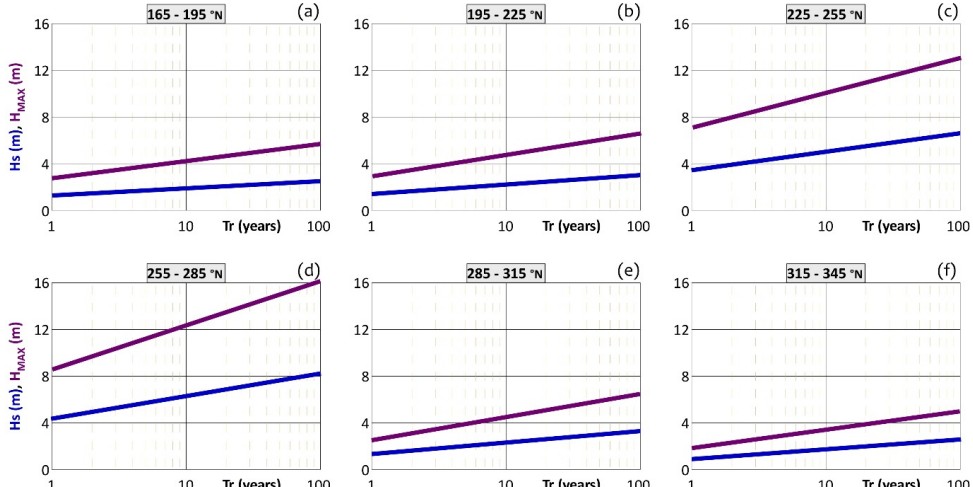

**Figure 9.** Significant (Hs, blue lines) and maximum (H$_{MAX}$, purple lines) wave height estimated by using the Gumbel distribution on the basis of the calibrated ERA5 dataset. Panels (a) – (f) illustrate the analysis for different wave directions.

### 3.2 Coupled model 2DEF + SWAN vs MIKE

The coupled model 2DEF+SWAN has been compared to MIKE during the storm event of 25 – 30 December 1999. Specifically, we selected the two sections located within the submerged barrier (i.e., sections 1 and 2, see Fig. 4). We

compared the significant wave height, which is affected by the local morphology, and the sea level, which is affected by the wind-wave setup (δ), from 00CET of 26 December to 00CET of 31 December 1999 (Fig. 10).


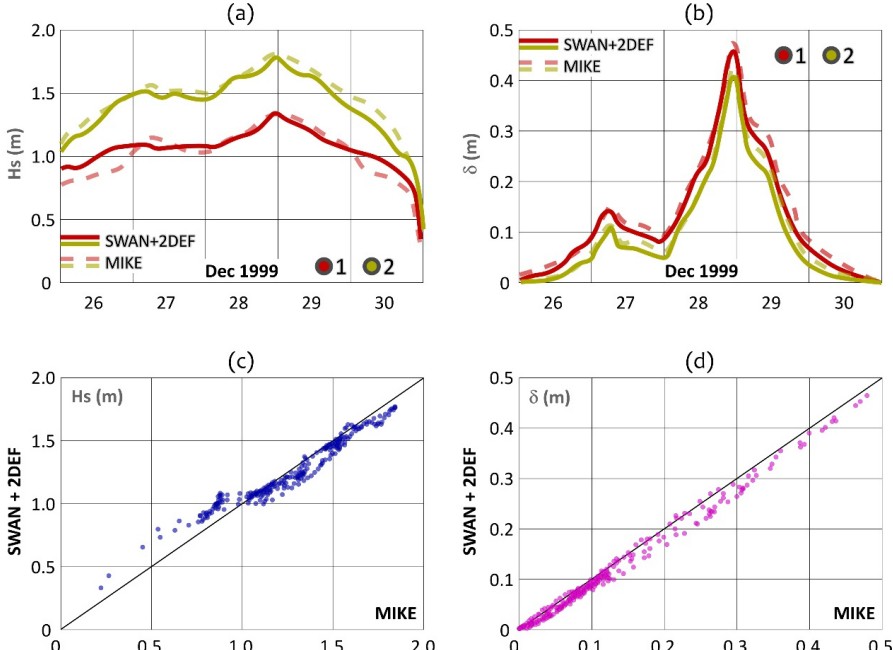

**Figure 10.** Event of 25 – 30 December 1999. Comparison between SWAN+2DEF and MIKE significant wave height ((a) and (c)) and wind-wave setup δ ((b) and (d)). (a) and (b) compare the two parameters over the time series 26 – 30 December for section 1 (red) and

section 2 (yellow, see Fig. 4 for the location of the sections); (c) and (d) show the scatter plots of the sections 1 and 2 aggregated data.

Results show a very good agreement between the two modeling systems, with a regression coefficient $R^2 > 0.99$ and a Fisher statistic F > 50,000 for both significant wave height (Hs) and wind-wave setup (δ). Standard error is < 0.1 m for Hs and < 0.01 m for δ. Notably, the computational cost of the coupled system SWAN+2DEF is significantly lower, as it needs less than 20 % of the time to run the same simulation by means of MIKE. This can be ascribed to the solution time step of the

wave climate by the SWAN module, (i.e., half hour), as MIKE solve both the wave climate and hydrodynamic every computational time step (i.e., few seconds).

### 3.3 Inshore wave propagation

The seabed topography significantly influences the inshore wave propagation by affecting the sea level and the wave celerity, height and direction approaching the shoreline. Specifically, the physical behavior of the waves propagating to

shallow areas is exhibited by wave shoaling, refraction, and breaking, tending the waves to become normal to the shoreline and to reduce their height, speed, and length, while wave period remains constant (Kirby and Dalrymple, 1994; Masselink





and Puleo, 2006; López-Ruiz et al., 2015; Joevivek et al., 2019). Figure 11 shows the inshore change in sea level (panel (a)), wave height (panel (b)), and wave direction (panel (c)) at Calabaia beach for different offshore wave directions and for a significant wave height Hs0 = 6 m. Results are compared at the five sections illustrated in Fig. 4. Our findings highlight a

significant difference in the three parameters when approaching the shoreline, with particular reference to the comparison between the sections located seaward the submerged breakwater (i.e., sections 3, 4, and 5) and sections 1 and 2. Wave height and setup peak when the offshore wave direction is almost perpendicular to the shoreline (i.e., 255 °N).

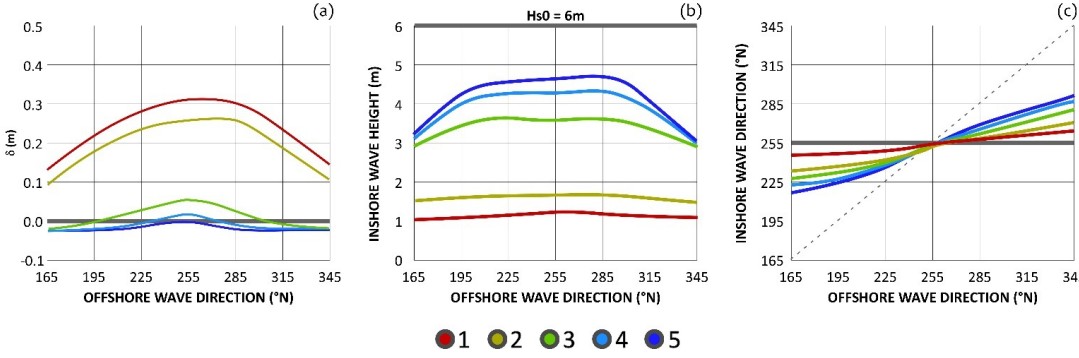


**Figure 11.** Wave shoaling, refraction and breaking effect on the sea level and wave characteristics approaching the shoreline of Calabaia for an offshore significant wave height of 6 m. (a) wave setup, (b) significant wave height reduction, (c) wave direction change.

### 3.3 Sea hazard projections

Figure 12 illustrates the flooding maps at Calabaia beach simulated by means of the coupled SWAN+2DEF model, with and

without the sea defense intervention (i.e., with SDI and no SDI) built in 2006, and for three different SLR scenarios (+ 0 m, + 1 m, and + 2 m). Results show the efficiency of the intervention in protecting the Calabaia settlement during the storm occurred on 28 December 1999, with particular reference to a SLR scenario of + 1 m (comparison between panels (b) and (e)), where the sea waves would not approach the village in case of SDI. Notably, a reduced distance between the shoreline and the urban settlement mey pose a significant threat for the village, as runup phenomena and foundation scour can damage

the building located in front of the sea.


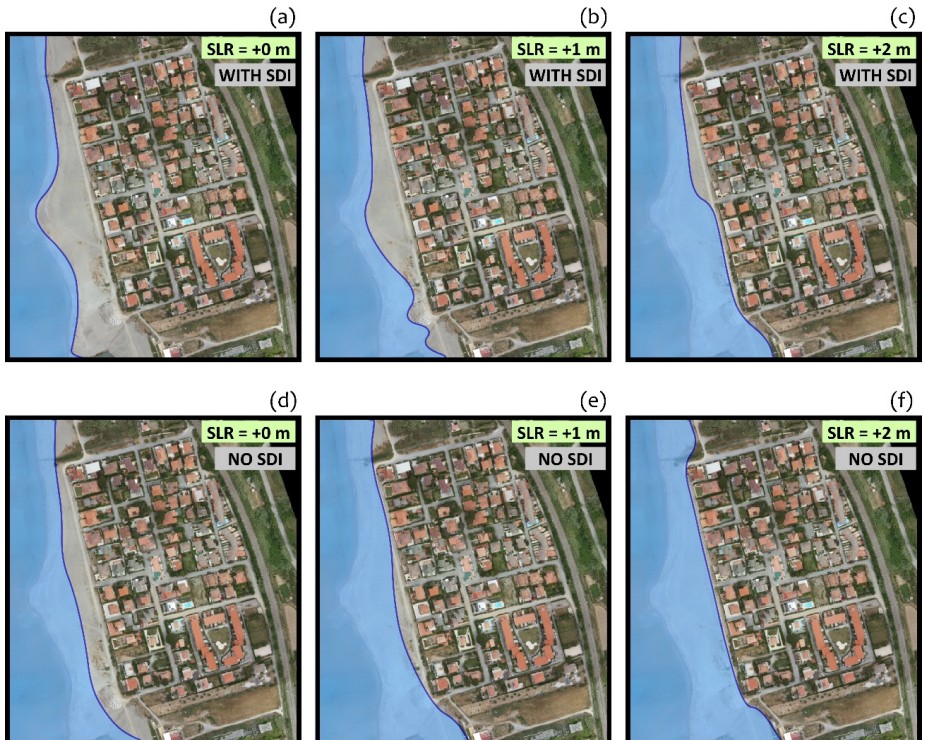

**Figure 12.** Event of 28 December 1999, 22 CET. Flooding maps of the Calabaia beach under different SLR scenarios (+ 0 m (a, d), +1 m (b, e), and + 2 m (c, f)). Panels (a) – (c) refer to the actual condition with the sea defense intervention (WITH SDI), panels (d) – (f) to the former (i.e., pre-2006) morphologic scenario, without any intervention (NO SDI).

For the same storm, we addressed the effect of SLR on the inshore wave height (Fig. 13a) and wave setup (Fig. 13b) at the two sections located within the submerged barrier (i.e., sections 1 and 2, see Fig. 4). A SLR of + 1 m would increase the inshore wave height of almost one third in both the sections, a SLR of + 2 m of almost two thirds (Fig. 13a). Conversely, the SLR would slightly reduce the inshore wave setup (Fig. 13b). Notably, similar differences in wave setup are also noticed if sections 1 and 2 are shifted closer to the shoreline, in order to achieve the same flow depth in the three SLR scenarios.


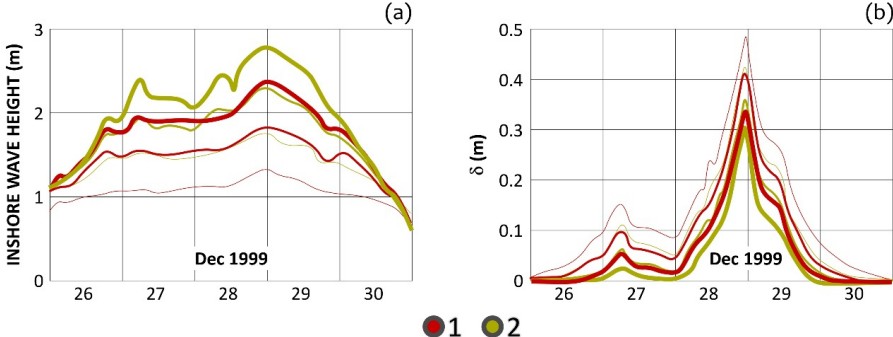

**Figure 13.** Event of 25 – 30 December 1999. Inshore significant wave height (a) and wave setup (b) computed at sections 1 (red) and 2 yellow (Fig. 4) for SLR scenarios of + 0 m (thinner lines), + 1 m (intermediate lines), and + 2 m (thicker lines).

The effect of the SDI on the inshore hydrodynamics is significant, affecting the longshore current and reducing the

longitudinal flow rate close to the shoreline. Figure 14 compares the inshore velocity field for a steady offshore wave of 4 m from 195 °N. SLR Scenarios of + 0 m and + 1 m have been analyzed comparing the condition with sea defense intervention (SDI), with beach nourishment only and no submerged barrier (NB), and with no intervention at all (NO). Results evidence the impact of the submerged barrier and beach nourishment in reducing the longshore current. This effect is fundamental in preventing further erosion processes on the shoreline, which can pose an additional threat for the Calabaia settlement.

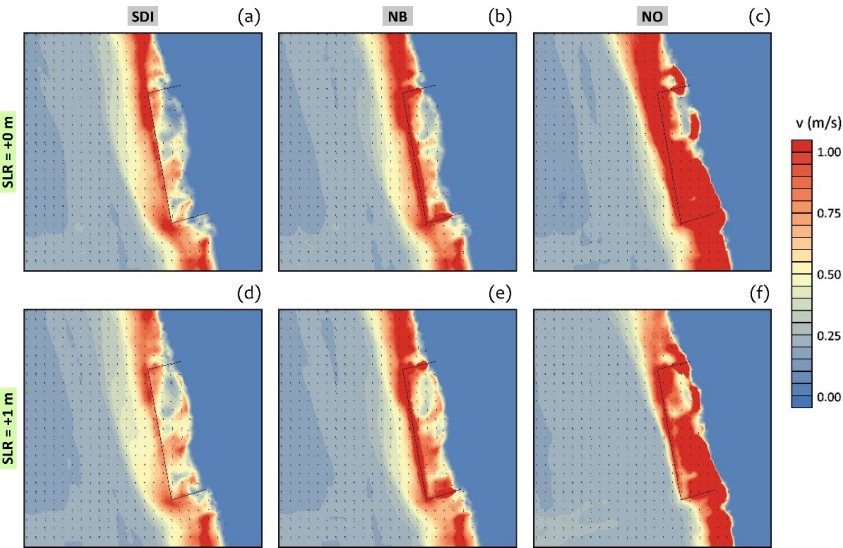


**Figure 14.** Offshore significant wave height of 4 m, incoming direction 195 °N. The panels compare the two-dimensional velocity field between the condition of submerged barrier plus beach nourishment (panels (a) and (d)); beach nourishment only (panels (b) and (e)); no intervention at all (panels (c) and (f)). (a) – (c) SLR scenario of + 0 m; (d) – (f) SLR scenario of + 1 m.

The effects of the SDI on the inshore hydrodynamic have been addressed in terms of the alteration of flow rates through the
transect $Q_A – Q_B$ (Fig. 15). We considered three offshore wave heights (Hs0, i.e., 4 m, 6 m, and 8 m), seven wave directions (i.e., 165 °N, 195 °N, 225 °N, 255 °N, 285 °N, 315 °N, and 345 °N), and two SLR scenarios (+ 0 m and + 1 m). Panels (a) and (b) respectively compare the actual fluxes ($Q_{SDI}$, i.e., with the SDI) to the conditions with no barrier ($Q_{NB}$) and no intervention at all ($Q_{NO}$). The comparison evidence that the SDI reduces the longshore flow rate through the transect $Q_A – Q_B$ for all the offshore wave characteristics and for both the SLR scenarios. The relationship is linear, with regression slopes of
1.45 (effect of the submerged barrier, panel (a)), and 2.75 (effect of the whole SDI, panel (b)). This wider difference is due to the significant change in the seabed elevation occurred after the beach nourishment. Figure 15c compares all the outcomings between the two SLR scenarios. Results show a flow rate increment of about 60 % for the + 1 m scenario, independently of the wave forcing and of the type of sea defense intervention. Figure 15d addresses the effect of the offshore wave height, indicating a flow rate increment, with respect to Hs0 = 4 m, of about 30% and 50% for Hs0 = 6 m and 8 m respectively.
Notably, all the relationships are linear, with a correlation coefficient $R^2$ greater than 0.95.

$$Q_{AB} \ (m^3/s)$$


**Figure 15.** Comparison of the flow rate through the transect $Q_A - Q_B$ (Fig. 4). The dataset includes all the wave heights and directions we simulated in steady conditions and the two SLR scenarios of + 0 m and + 1 m. (a) condition with sea defense intervention vs beach nourishment only; (b) condition with sea defense intervention vs no intervention at all; (c) sea level rise of + 0 m vs + 1 m; (d) significant offshore wave height of 4 m vs 6m (red) and 8 m (yellow).

## 4 Conclusions

The enhancing storm impact on coastal areas have reshaped the history of many urban settlements and communities, with wave erosion and flooding causing widespread devastation. However, the lack of awareness of the climate change effect on the possible occurrence of more hazardous events in the intervening decades has often resulted in an overreliance of former sea defenses or a loss of folk-memory (Hansom et al., 2015). In this context, structural measures, even if combined with high level of technical knowledge, are not the panacea for the long-term safety of coastal areas (Mel, 2021). Adaptation to climate change requires an integrated knowledge and management, into a strategic plan that includes the social, economic, and environmental issues of coastal areas Allocating resources to the development of mathematical models is indeed crucial to estimate further effects of climate change and sea defense interventions on the vulnerability of coastal areas.

Toward this goal, in this study, the inshore wave climate and hydrodynamic have been analyzed at Calabaia beach (Italy). The analyses have been supported by the ERA 5 hindcast (1950 – 2019) and buoy (1999 – 2008) dataset, together with high-resolution topographic and bathymetric surveys. Specifically, a coupled wave-current numerical model system was developed by using SWAN wave model and 2DEF hydrodynamic model. Both the uncoupled models perform well in reproducing measured wave and hydrodynamic parameters (Booij et al., 1996; Booij et al., 1999; Defina, 2003; Mel et al., 2021). The coupled system improves the performance of the simulation with respect to the uncoupled system. Furthermore, the outcomes are similar to other commercial models, but with a significantly lower computational cost. The coupling consists of feeding the wave model with bottom elevation, sea level, and surface currents computed by the hydrodynamic model and returning the radiation stress and the orbital wave velocity to the latter. Fields were exchanged half-hourly between the two modules.

We analyzed different scenarios of SLR and wave climate with and without the sea defense intervention built in 2006. Simulations were carried out considering both synthetic and realistic offshore wave forcings. Nearshore modelling has provided instructive insight into the spatial variability of wave climate, such as the effect of wave shoaling, refraction and breaking, and the efficiency of the sea defence intervention. Specifically, the absence of such measure, together with se level rise, would increase the vulnerability of the Calabaia shoreline, with particular reference to the wave height and longshore currents. Furthermore, we found multiple linear relationships of the effects on the longshore flux of the (i) sea defence intervention; (ii) sea level rise; (iii) and wave climate.

The results of this study can be useful to design further sea defense structures, to support analysis in restoring or designing new offshore structures, and to provide more insights on coastal erosion in the southern Tyrrhenian shoreline. New data and



monitoring systems will be soon achieved in the area, such as the possibility to repeat the UAV flights periodically, allowing
to achieve an extremely precise monitoring of the coastline. Calabaia beach pertains to the Marine Experimental Station of
Capo Tirone, which is set to become a crucial hub for supporting a sustainable development of the southern Tyrrhenian
coastline. Overall, the present study is held to be of general interest for multiple research purposes, such as (i) the
development and test of high-resolution coupled modelling systems in coastal areas; (ii) their implementation in operational
oceanography platforms; (iii) coastal planning and management support, assessing the potential hazard posed by extreme
storms together with the efficiency of sea defence interventions; (iv) production of long-term scenarios in view of climate
change, with particular reference to the hazard assessment related to climate change. Future studies will focus on the
development of an additional morphodynamic module to estimate the long-term erosion process and the shoreline evolution.
We note that sea level rise and storminess variability may result in a nonlinear response of the landscape. Furthermore, new
coastal interventions will require to balance the flooding protection with the economic interests and safeguarding of the
whole coastal ecosystem. In this context, it is crucial a long-term planning, which should focus on the restoration of the
environment and on the sustainability and durability of different measures, considering all the needs of the land, in accord
with the most recent EU directives, as coastal areas are a *unicum* environment that does not obey to any national or decision-
making boundary.

## Funding

This research is partially funded by the Italian Ministry of University and Research (MIUR) through the project PON-AIM
"Attraction and International Mobility", Action I.2, CUP: H24I19000360005.Calabria.

## Acknowledgements

Bruno Matticchio and Devis Canesso are gratefully acknowledged for their support in implementing the SWAN+2DEF
coupled modelling system.

**Author contributions**

R.A. Mel: conceptualization, methodology, investigation, software, validation, visualization, writing original draft, writing
review & editing; T. Lo Feudo: conceptualization, data curation, validation, visualization; M. Miceli: data curation,
validation, writing review & editing; S. Sinopoli: conceptualization, methodology, investigation, software, validation,
visualization; M. Maiolo: conceptualization, methodology, validation, visualization, writing review & editing, supervision,
project administration.



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
