# Peer review of "A coupled modelling system to assess the effect of Mediterranean storms under climate change"

_Natural Hazards and Earth System Sciences, 2022_

## Author Comment (AC1)

**RESPONSE TO REVIEWERS**

Please note that in this rebuttal, *italics* refer to the text of the reviewers' comments, our detailed response is in black, the new text of the revised version is in **bold blue**.

**REVIEWER #1:**

**GENERAL COMMENT:**

*This work describes the first steps towards an innovative fully coupled modelling system composed of a hydrodynamic (2DEF) and wind wave model (SWAN). Numerical simulations have been performed to identify the impact of extreme storms at Calabaia beach by combining sea level rise and extreme wave projections with the most recent georeferenced territorial data both in a case study and in a climatological perspective. The paper is appropriate for publication in NHESS but it requires major revisions.*

We thank the reviewer for his/her constructive criticism. We seriously considered his/her suggestions, amended some points, and further improved the manuscript. Detailed answers are reported below.

**MAIOR POINTS:**

1) *The title "A coupled modelling system to assess the effect of Mediterranean storms under climate change" is extremely vague. It may refer to meteorological, ocean, wave models so you should clarify already in the title its specific focus on coupled hydrodynamical and wave models.*

   The reviewer is right. In the new version of the manuscript, we changed the title in:

   > **A coupled wave-hydrodynamical model to assess the effect of Mediterranean storms under climate change**

2) *The abstract should focus on the results of the paper. Its first part is very generic and can be applied to any work in the field. There is no indication on the application of the model to a case study and to the climatology, which is not appropriate.*

   We agree with the reviewer. In the new version of the manuscript, we rewrote the abstract as follows:

   **Climate change will have an undeniable influence on coastal areas, resulting in increased rates of both sea level rise and storm-related impacts. In this context, it is crucial to estimate the local probable extreme sea wave conditions, to properly reproduce the sea state and the coastal hydrodynamic, and to investigate the effectiveness of sea defenses under sea level rise. This work describes the first steps towards an innovative fully coupled modelling system composed of a wind- sea wave (SWAN) and hydrodynamic model (2DEF). Numerical simulations, focusing on Calabaia beach, Italy, have been compared to the MIKE outcomes in the same area. The simulations have been performed to study the inshore sea wave characteristics, to assess the effectiveness of the actual sea defence interventions, and to identify the impact of extreme storms, by combining sea level rise and extreme sea wave scenarios with the most recent georeferenced territorial data. The models are two-way coupled at half-hourly intervals exchanging the following fields: 2D sea level, surface currents and bottom elevation are transferred from 2DEF to SWAN; sea wave**

**characteristics computed by SWAN is then passed to 2DEF by modifying the radiation stress.**

3) *Line 201: what do you mean with "wave climate"? here, you are not referring to a wave climatology and to different possible wave directions. Please, clarify.*

In the new version of the manuscript, we refer wave height, period and direction to "wave characteristics" and wave climatology to "wave climate". We corrected all these terms thorough the manuscript accordingly. Thank you for noting.

4) *Line 244-245: the meteorological description is confusing, for example: a strong divergence area may be relevant for cyclone development, but divergence does not originate cyclones; extra-tropical cyclones have diameters of about 1000 km, 300 km is not realistic at all. Line 251: "the medium wind speeds were within the range expected for a strong storm": sorry, but I do not understand what you mean here.*

In the new version of the manuscript, we slightly reduced the meteorological description by deleting the possible confusing sentences.

5) *Lines 267-287: this part provides unnecessary results, since they are well known and references would be sufficient; Figure 7 is not necessary, but if you want to include it, you should ask the permission for reproduction.*

Section 2.6 has been significantly reduced. The source of data of Figure 7 has been mentioned in the caption.

6) *Section 3: The entire section should be rearranged, distinguishing the case study analysis from the climatological results. Now you go back and forth, and the two parts are not clearly distinguished. I think that separating the two different analyses, the one referring to the case study from that considering the synthetic (climatological) analysis would strongly improve the readability of the paper.*

In the new version of the manuscript, we improved the readability of the whole paper, by moving/rewriting some part of the text. Thank you for the advice.

7) *Figures: Figure 8: what is the interpretation of the squares in panel a? Figure 9: the figure is not commented on in the text.*

In the new version of the manuscript, we fixed both the issues. Thank you for noting.

8) *Conclusions: I do not think you show in any place that "The coupled system improves the performance of the simulation with respect to the uncoupled system".*

In the new version of the manuscript, we deleted this misleading sentence.

9) *Line 63: change "the Mediterranean cyclone" into "the presence of Mediterranean cyclones …"*

10) *Line 66: change into "… makes the Mediterranean sea subject to …"*

11) *Line 67: … southerly winds …*

12) *Line 68: … easterly winds …*

13) *Line 99: … affecting the …*

14) *Line 102: you mention that you are using ERA-INTERIM but in other points you indicate ERA-5. Please clarify!*

15) *Line 103: "The most important recent storm which affected the Mediterranean Sea (25 – 29 December 1999)": several intense storms affected the Mediterranean Sea in the last 20 years. It is not clear why this should be the most important.*

16) *Line 123: "the eastern, southern and eastern sides": please correct;*

17) *Line 126: I do not understand why you refer here to a meteorological paper while you are describing geographical features;*

18) *Line 133: defenses -> defences;*

19) *Line 139: third and fourth quadrant: you should clarify which direction you are referring to;*

20) *Line 146 and elsewhere: statistics instead of statistic and it is plural (e.g., statistics have …);*

21) *Line 197: "The model grid is closed 4 km north of Diamante, 3 km south of Cape Bonifati …": either you should report the places on the map or you should not mention them in the text;*

22) *Line 420: again, please clarify what you mean with wave climate;*

23) *Line 450: change into "A long term planning is crucial"..*

In the new version of the manuscript, we fixed all the reported minor issues.

In particular:

- As concerns issues 15, we added a reference (**Ulbrich et al., 2001**).

- As concerns issue 17, we summarized the first paragraph of Section 2.1 in:

**Calabria region is in the south of Italy, ranging between 37°55' and 40° latitude North and between 15° 30' and 17° 15' longitude East. The western part of the region is bounded by the southern Tyrrhenian Sea, while the southern, and eastern sides are bounded by the Ionian Sea (Fig. 1).**

---

## Author Comment (AC2)

**RESPONSE TO REVIEWERS**

Please note that in this rebuttal, *italics* refer to the text of the reviewers' comments, our detailed response is in black, the new text of the revised version is in **bold blue**.

**REVIEWER #2:**

GENERAL COMMENT:

*The article presents and application of flow of wave modelling nearshore Calabaia considering different sea level rise values and offshore wave conditions. I find the application shown in the article interesting but the validity of the applied models and the rationale behind the considered conditions lacking. As a whole the article needs restructuring, proof- and critical-reading. There are a few buzzwords being applied in a non-coherent way. For instance, what is the meaning of "global risk society" and "Climate change drives potential future sea hazards, as the greenhouse effect is expected to lead to global warming"? Furthermore, a lot of what is stated in the abstract and introduction can be removed as they are not directly related nor motivating the contents. I suggest that the authors rewrite the whole article. Below a few questions and suggestions. The list is not comprehensive, but at least these issues should be addressed before the article can be assessed.*

We thank the reviewer for his/her constructive criticism. We seriously considered his/her suggestions, amended some points, and further improved the manuscript. Some parts of the paper have been rewritten and/or moved to improve the whole readability. Detailed answers are reported below.

1) *Line 14: What is the innovative aspect? Not the coupling between waves and hydrodynamics nor considering the effects of SLR on coastal waves and loads, for these you can find many references.*

   The innovative aspect is the coupling of 2DEF and SWAN models. This coupling system optimize the computational cost by running SWAN at the same time step of the boundary condition data.

2) *Lines 17-18: If you are considering projections then it should be the "projected impact" or "impact projections".*

   Fixed, thank you for noting.

3) *Replace "Sea waves, caused by the effect of local wind climate" with "Sea or wind waves, forced by the local winds".*

   Fixed, thank you.

4) *What do you mean with "Interaction between sea and swell waves can cause unpredictably high waves"? How and why is it unpredictable?*

   In the new version of the manuscript, we deleted this confusing sentences.

5) *Can "Extreme sea stormy conditions" be replaced with "Coastal extreme storms"?*

Yes, thank you.

6) *Lines 36-38: Please clearly state what you mean with: "sea storms", "sea waves" and "swell waves". How do you refer to (sea+swell) wave conditions?*

In the new version of the manuscript (lines 23 – 25) we clearly stated the meaning of the three terms:

**Hereinafter, we refer to wind waves the waves triggered by the local winds, to swell waves the waves moving inshore from a distance, and to waves (i.e., sea waves) the superimposition of wind and to swell waves.**

Thank you for the advice.

7) *Lines 76 to 83 can be removed as they do not contribute to the subject.*

Fixed. Thank you.

8) *Line 95: "Sea level and surface currents are driven by sea state," this is not true. Please rephase or expand acknowledging that sea levels and currents are driven by atmospheric and astronomical forcing.*

Fixed. Thank you.

9) *Lines 99-101: Please rephrase of expand. The Gumbel distribution could in principle be used to model the annual maxima, not to "identify the extreme sea wave conditions" By "sea wave" you mean only wind sea?*

Fixed. Thank you.

10) *Lines 101-102: I assume that you mean ERA5 and not ERA-interim as it does not cover the full period given. Assuming that it should be ERA5, the quality of the ERA5 data is expected to be lower in the 1950-1979 period. Have you checked whether there are inhomogeneities in the ERA5 data before and after 1979?*

Thank you for noting the misleading term used in the former version of the manuscript. The homogenesis of the data has been checked in our former work (Lo Feudo et al., 2022), cited in the manuscript.

11) *Line 140-141: Define "wave currents". Are they only in the wave break-zone? Do they exclude direct wind and pressure forcing?*

In the new version of the manuscript, we wrote (lines 116 – 117):

**…with a negligible interaction between tidal currents and longshore wave currents.**

12) *Line 147: The ERA5 data are available hourly, why do you only consider 6-hourly?*

For climate studies, a 6-hourly dataset is the best compromise to produce a reliable climate analysis and the storage-cost.

13) *Lines 201-202: What do you mean with "model is forced by imposing (refer to Fig. 3a): (i) wave climate at the seaward boundary section (yellow dashed line) every 30 minutes". By climate I understand long-term means or return values and these are generally assumed stationary, not changing every 30 minutes.*

In the new version of the manuscript, we refer wave height, period and direction to "wave characteristics" and wave climatology to "wave climate". We corrected all these terms thorough the manuscript accordingly. Thank you for noting.

14) *Lines 144-146: I find the description of Lothar incorrect. Can you please check your references or https://en.wikipedia.org/wiki/Cyclone_Lothar?*

We checked but unfortunately we did not find any misleading description of Lothar.

15) *Line 258: Why "CFS reanalysis" and not ERA5, as you are using ERA5 in the study?*

In the new version of the manuscript, we replaced CFS with ERA5 reanalysis panels. Thank you for the advice.

16) *Section 2.6 should be significantly reduced, most of it is copied from the IPCC report. Just refer to it and state the values that will be considered further.*

In the new version of the manuscript, Section 2.6 has been significantly reduced.

17) *Lines 245-247: Please rewrite. It is not clear from the text that the models use different numeric schemes and which. Is also confusing to be faced again with "climate", what do you mean?*

In the new version of the manuscript, we better stated the model setup thorough the manuscript. Thank you for noting.

18) *Section 3.1: The return value estimates are not used further. Why are they presented? To motivate the range of values given in line 395?*

The reviewer is right. We presented the return values to motivate the further synthetic analysis.

19) *Section 3.2: In this section the 2DEF+SWAN results are validated by comparing them with the results of another model and none of the models are validated against observations. The validity of the model results is therefore not verified. The only conclusion/aim appears to be to show that the results are comparable and running times of 2DEF+SWAN lower.*

We agree with the reviewer. However, the case study is, one hand site-specific. On the other hand, since at Calabaia Beach the morphology is very simple, this case study can have general application. Moreover, both SWAN and 2DEF (and MIKE) models and have been widely tested over decades (we provided some references thorough the manuscript). Finally, we highlight that the target of this work is to compare the two methodologies (SWAN+2DEF and MIKE modelling systems), ensuring that the results provided are similar on order to verify the effectiveness of the coupling system.

*20) Section 3.3: There are two sections 3.3. Please correct.*

Fixed, thank you for noting.

---

## Author Comment (AC3)

**RESPONSE TO REVIEWERS**

Please note that in this rebuttal, *italics* refer to the text of the reviewers' comments, our detailed response is in black, the new text of the revised version is in **bold blue**.

**REVIEWER #3:**

**GENERAL COMMENT:**

*A coupled hydrodynamic (2DEF) and wave (SWAN) model is presented. The model is tested for storms in Calabaia beach by including sea level rise and extreme wave projections. The coupled model is assessed against results from Mike model.*

*One of my main criticisms to this work is that the results of the coupled hydrodynamic and wave model are evaluated against another model (Mike). Throughout the whole text the English expressions need to be reviewed and corrected and representation of results and figures need to be improved. I also suggest editing the title to be more specific.*

We thank the reviewer for his/her constructive criticism. We seriously considered his/her suggestions, amended some points, and further improved the manuscript. Some parts of the paper have been rewritten and/or moved to improve the whole readability. Detailed answers are reported below.

We agree with the reviewer that the validity of the results of the proposed model has been verified against another model only (i.e., MIKE). However, on one hand, the case study is site-specific. On the other hand, since at Calabaia Beach the morphology is very simple, this case study can have general application. Moreover, both SWAN and 2DEF (and MIKE) models and have been widely tested over decades (we provided some references thorough the manuscript). Finally, we highlight that the target of this work is to compare the two methodologies (SWAN+2DEF and MIKE modelling systems), ensuring that the results provided are similar on order to verify the effectiveness of the proposed coupled system.

1) *Abstract: The abstract is so general and does not include that Mike model is used for assessing the couple hydrodynamic and wave model. Line 12: Please explain what you mean by "wave and hydrodynamic inshore field".*

   We agree with the reviewer. In the new version of the manuscript, we rewrote the abstract as follows:

   **Climate change will have an undeniable influence on coastal areas, resulting in increased rates of both sea level rise and storm-related impacts. In this context, it is crucial to estimate the local probable extreme sea wave conditions, to properly reproduce the sea state and the coastal hydrodynamic, and to investigate the effectiveness of sea defenses under sea level rise. This work describes the first steps towards an innovative fully coupled modelling system composed of a wind- sea wave (SWAN) and hydrodynamic model (2DEF). Numerical simulations, focusing on Calabaia beach, Italy, have been compared to the MIKE outcomes in the same area. The simulations have been performed to study the inshore sea wave characteristics, to assess the effectiveness of the actual sea defence interventions, and to identify the impact of extreme storms, by combining sea level rise and extreme sea wave scenarios with the most recent georeferenced territorial data. The models are two-way coupled at half-hourly intervals exchanging the following fields: 2D sea level,**

**surface currents and bottom elevation are transferred from 2DEF to SWAN; sea wave characteristics computed by SWAN is then passed to 2DEF by modifying the radiation stress.**

2) *Introduction: The introduction is about three pages long and most of the sentences are off topic. It does not provide enough information about the topic of the paper: coupled modelling. I suggest rewriting this section while focusing on relevant literature and putting them in the context of this study.*

We removed most of the sentences off topic and we restructured the whole introduction, thank you.

3) *Line 21: I suggest rewording the sentence: "Coastal areas contain a wide amount of life"*

Fixed, thank you.

4) *Line 39: I suggest rewording the sentence: "sea stormy conditions"*

Fixed, thank you.

5) *Line 58 & 59: "This is particularly significant in case of micro-tidal environments, such as the Mediterranean Sea, where extreme events are expected to be superimposed to SLR scenarios, exacerbating the flooding hazard even in the case of a possible storminess reduction. Please explain the logic of this sentence. The word "micro-tidal" refers to small tidal range, how can that be a reason for "exacerbating the flooding hazard"*

In small tidal range environments, 20-30 cm can be significant (with respect to the tide amplitude) in determining the flooding of such areas. For this reason, SLR is particularly hazardous.

6) *Line 84: Please explain what you mean by "the design of maritime works"*

As we summarized the introduction, in the new version of the manuscript, such sentence is not reported anymore.

7) *Line 103: "The most important recent storm"Please explain what makes the storm "the most important".*

In the new version of the manuscript, we added the reference **Ulbrich et al., 2001**. Moreover, the characteristics/importance of the storm are widely described/highlighted in Section 2.5.

8) *Line 124 to 126: "Apennines run along the whole region from north to south, consisting of five main ranges, namely, Pollino, Catena Costiera, Sila, Serre, and Aspromonte, characterized by peaks heights between 1,500 m and 2,000 m (Federico and Bellecci, 2004)." Please explain how this study is related to Apennine Mountains?!*

We summarized the first paragraph of Section 2.1 in:

**Calabria region is in the south of Italy, ranging between 37°55' and 40° latitude North and between 15° 30' and 17° 15' longitude East. The western part of the region is bounded by the southern Tyrrhenian Sea, while the southern, and eastern sides are bounded by the Ionian Sea (Fig. 1).**

9) *Line 146 & 147: Here you are referring to ERA, while in Line 102 you have referred to ERA-Interim. Which one has been used in this study? The temporal resolution of ERA5 reanalysis is 1 hour, please explain why 6-hour resolution is used in this study. Also add information about the spatial resolution.*

We used ERA5 and we fixed the misleading term, thank you.

We note that for climate studies, a 6-hourly dataset is the best compromise to produce a reliable climate analysis and the storage-cost. More details about the data can be found in **Lo Feudo et al., 2022.**, mentioned in such paragraph.

10) *Line 151 to 153: "Although the actual performance of wind-wave models is generally good, for closed basins (i.e., the Mediterranean Sea) winds forcing is generally underestimated, with a significant impact on wave modelling due to the lack of knowledge of detailed physiographic features." Please provide references confirming this.*

In the new version of the manuscript, we added in that sentence the reference **Cavaleri and Bertotti, 2004**.

11) *Line 154: "ECMWF wind fields" Is this ECMWF ERA5 reanalysis?*

Yes, thank you for noting.

12) *Lines 153 to 157: Considering that ERA wave data are available in 0.5-degree spatial resolution, please explain how the ERA5 data are matched with the buoy data. Also, explain the possible reason behind differences between the two sources?*

We refer **Lo Feudo et al., 2022** for the analysis and comparison between the two sources of data.

13) *Figure 2: Please explain the horizontal scatter of points in panel b where mean periods from ERA5 are around 3s*

Very-few data show this behaviour in calm sea condition, possibly due to boat waves (of negligible height) that affect the recorded signal.

14) *Line 172: Please explain why 2DEF model is used. Defina 2003 is about "Numerical Experiments on Bar Growth". How has that been the best model for hydrodynamics here?*

2DEF model has been widely applied in lagoonal areas. In the present manuscript, we coupled the 2DEF model with SWAN in order to solve the complete wave field (see lines 149 -160).

15) *Line 188: "The SWAN grid is included into the boundaries of the 2DEF domain." This sentence is unclear.*

In the new version of the manuscript, we wrote:

…the SWAN **domain** is included into the boundaries of the 2DEF domain.

*16) Line 197: "The model grid is closed 4 km north of Diamante" The sentence is unclear.*

In the new version of the manuscript, we removed this misleading sentence. Thank you for noting.

*17) Lines 205 & 206: "Specifically, we reproduced seven wave directions (i.e., 165 °N; 195 °N; 225 °N; 255 °N; 285 °N; 315 °N; 345 °N) and, for each wave direction, three significant wave heights (i.e., 4 m; 6 m; 8 m)." This sentence is very confusing! What does it mean to reproduce 3 Hs for each wave direction?!*

The reviewer is right. In the new version of the manuscript, we wrote (lines 183 – 188):

**Specifically, we reproduced seven wave directions (i.e., 165 °N; 195 °N; 225 °N; 255 °N; 285 °N; 315 °N; 345 °N) and, for each wave direction, three significant wave heights (i.e., 4 m; 6 m; 8 m). For each of the 1+7·3 = 22 simulations, we reproduced five scenarios of sea level rise (i.e., +0.0 m; + 0.5 m; +1.0 m; +1.5 m; +2.0 m) and, in turn, three morphological scenarios (i.e., present condition of the shoreline, after the intervention described in Section 2.1 and in Maiolo et al., 2020a and 2020b; former condition, without such intervention; present condition without the submerged barrier and the semi-submerged groynes), for a total of 22·5·3 = 330 simulations.**

*18) Line 246: "hurricanes of category 2" Please explain what that means.*

In the new version of the manuscript, we added the following reference: **Klotzbach et al. 2020**.

*19) Figure 5: Where is the study area in this figure?*

We fixed this issues in the new panels, which refer to ERA5 reanalysis. Thank you for the advice.

*20) Figure 6: I suggest separating the wave height, period, and direction into 3 different subplots.*

Fixed, thank you for noting.

*21) Line 303: "Extreme wave climate study is based on data collection, selection, and analysis" This sentence seems to be off topic.*

We removed it, thank you for noting.

*22) Line 412: "The enhancing storm impact on coastal areas have reshaped the history of many urban settlements and communities". Please rewrite this sentence with better choices of words.*

We rewrote the first sentence of the conclusion as:

**Flooding and wave erosion driven by climate change has reshaped the history of many coastal settlements and communities. However, the lack of awareness of the climate change effect on the possible occurrence of more hazardous events in the intervening decades has often resulted in an overreliance of former sea defenses or a loss of folk-memory (Hansom et al., 2015). In this context, structural measures, even if combined with high level of technical knowledge, are not the panacea for the long-term safety of coastal areas (Mel, 2021).**

23) *Line 426: "The coupled system improves the performance of the simulation with respect to the uncoupled system. Furthermore, the outcomes are similar to other commercial models, but with a significantly lower computational cost." Where in the text have these been demonstrated? Please name which commercial models you are referring to.*

In the new version of the manuscript, we wrote (lines 393 – 394):

**The outcomes are similar to other commercial models (e.g., MIKE), but with a significantly lower computational cost.**

Thank you for noting.

24) *Major technical correction is needed in the text, on top please note that there are two 3.3 sections in the manuscript.*

Fixed, thank you for noting. All the manuscript has been checked by all the authors.